# Synthetic Environmentally Friendly Castor Oil Based-Polyurethane with Carbon Black as a Microphase Separation Promoter

**DOI:** 10.3390/polym11081333

**Published:** 2019-08-12

**Authors:** Jia-Wun Li, Wen-Chin Tsen, Chi-Hui Tsou, Maw-Cherng Suen, Chih-Wei Chiu

**Affiliations:** 1Department of Materials Science and Engineering, National Taiwan University of Science and Technology, Taipei 10607, Taiwan; 2Department of Fashion and Design, LEE-MING Institute of Technology, No. 22, Sec. 3, Tailin. Rd., New Taipei City 24305, Taiwan; 3Sichuan Provincial Key Lab of Process Equipment and Control, Material Corrosion and Protection Key Laboratory of Sichuan Province, College of Materials Science and Engineering, Sichuan University of Science and Engineering, Zigong 643000, China; 4Department of Fashion Business Administration, LEE-MING Institute of Technology, New Taipei City 24305, Taiwan

**Keywords:** carbon black, microphase separation promoter, polyurethane, vegetable oil, environmentally friendly

## Abstract

This study created water polyurethane (WPU) prepolymer by using isophorone diisocyanate, castor oil, dimethylolpropionic acid, and triethanolamine (TEA) as the hard segment, soft segment, hydrophilic group, and neutralizer, respectively. TEA, deionized water, and carbon black (CB) were added to the prepolymer under high-speed rotation to create an environmentally friendly vegetable-oil-based polyurethane. CB served as the fortifier and promoter of microphase separation. Fourier transform infrared spectroscopy was performed to elucidate the role of H-bond interactions within the CB/WPUs. Additionally, atomic force microscopy was conducted to determine the influence of H-bond interactions on the degree of microphase separation in the WPU. Furthermore, this study used four-point probe observation to discover the materials’ conductivity of CB in the WPU. Thermogravimetric analysis and dynamic mechanical analysis were performed to measure the thermal properties of the CB/WPUs. The mechanical properties of CB/WPUs were measured using a tensile testing machine. The CB/WPUs were also soaked in 1 wt.% NaOH solution for different amounts of time to determine the degradation properties of the CB/WPUs. Finally, scanning electron microscopy was performed to observe the topography of the CB/WPUs after degradation.

## 1. Introduction

The promotion of green materials has attracted global attention to the development of environmentally friendly polymers. Numerous studies have investigated vegetable oils [1,2], such as castor oil (CO) [3,4] and soybean oil [5]. CO is extracted from the seeds of caster plants. Because every CO molecule consists of three –OH groups, CO can serve as a cross-link agent [6]. Scholars have imported CO into polyurethane (PU) structures [7,8], and even implemented the postmodified ester [9,10] and alkenyl [11,12] groups of CO in PU. PU has outstanding mechanical properties, abrasion resistance, and stiffness [13,14]. Therefore, this study synthesized green PU by using natural materials and isocyanate [15]. Water polyurethane (WPU) has inferior mechanical properties 5 to oil PU [16]. The mechanical properties of PU are mainly determined by the degree of microphase separation during PU formation. Microphase separation is generated by the polarity between hard and soft segments, which also causes the clustering of hard segments [17,18]. The formation of microphase separation mainly arises from the generation of hydrogen bond between the -NH group on urethane group (–COONH–) hard segment and C=O group, which promotes the aggregation of hard segment.

Scholars have employed 1-octadecanol as a promoter to improve the degree of microphase separation and mechanical properties in PU [19]. In other studies, the H-bond interaction between inorganic substances and PU has been used to increase the degree of microphase separation. Xia et al. discovered that adding a suitable amount of silicate clay increased the degree of completion of microphase separation during the PU production process. However, the addition of excessive amounts of clay lowered the degree of microphase separation [20]. Studies have also reported that the addition of multiwalled nanotubes accelerated the microphase separation of structures within the thermoplastic PU matrix [21,22]. This type of microphase separation is mainly caused by the H-bond interaction formed by the –COOH or –OH group of the inorganic substance and the C=O group in the thermoplastic PU. The increase in the degree of microphase separation also improved the mechanical properties of PU. Although WPU has poorer mechanical properties than PU, the WPU synthesis process does not require the use of solvents. Thus, the production of WPU creates fewer volatile organic compounds and is thus more environmentally friendly [23]. In one study, scholars used isophorone diisocyanate (IPDI) and CO to successfully synthesize WPU with outstanding mechanical properties [24]. The present study aimed to synthesize the aforementioned type of green PU and employ inorganic substances as the microphase separation accelerator for producing environmentally friendly PU nanocomposites. The inclusion of inorganic substances in PU can endow PU with various functions [25,26]. For example, the blending of PU and Ag produced an antibacterial material [27]; the use of oyster shell powder as a nucleating agent in PU imbued PU with the antibacterial properties of the oyster shell powder, producing a biomedical material [28]; and the addition of carbon material to PU produced a conductive material [29]. Carbon black (CB) is a natural product produced when hydrocarbon substances are burned in insufficient air. The surface layer of CB comprises –OH groups, which can form H-bond interactions with C=O groups in PU. These groups not only can attract the hard segments to around CB to increase the microphase separation, but also can promote the spread of CB within PU, creating PU with conductivity and adding to the functionality of PU to enable production of static conductive materials. This study employed the –OH group in CB to promote microphase separation in CO-based PU and endow PU with static conductive functionality.

This study employed IPDI, CO, dimethylolpropionic acid (DMPA), and triethanolamine (TEA) as the hard segment, soft segment, hydrophilic group, and neutralizer, respectively, to create a WPU prepolymer. Scheme 1 displays the process of adding TEA, deionized water, and CB to the PU under high-speed rotation to produce environmentally friendly CO-based WPU. Figure 1 shows the cycle of environmental CB/WPUs. Fourier transform infrared spectroscopy (FTIR) was used to determine the functional group changes. Thermogravimetric analysis (TGA) and dynamic mechanical analysis (DMA) were used to investigate the thermal properties of the CB/WPUs. Additionally, the mechanical properties and surface topography of the CB/WPUs were obtained using a tensile testing machine and atomic force microscopy (AFM). A four-point probe was employed to measure the conductivity of the CB/WPUs. Subsequently, this study analyzed the degradation properties of the CB/WPUs after they were soaked in NaOH for different periods of time and employed scanning electron microscopy (SEM) to observe the topography of the hydrolyzed CB/WPUs.

## 2. Experimental

### 2.1. Materials

Methyl ethyl ketone (MEK) from Mallinckrodt Baker Chemical Inc. (Phillipsburg, NJ, USA) was dehydrated with 4-Å molecular sieves for several days before its use as a solvent. Dimethylolpropionic acid (DMPA) and triethylamine (TEA) were purchased from Aldrich Chemical Co. (Milwaukee, WI, USA). Isophorone diisocyanate (IPDI) was obtained from Tokyo Chemical Industry Co. Ltd. (Tokyo, Japan). Ethylenediamine (EDA) was obtained from Tedia Company Inc. (Fairfield, OH, USA). Castor oil (CO) was purchase from Alfa Aesar (Ward Hill, MA, USA). Carbon black was from Lingo Industry Co., Ltd. (Jhongli, Taiwan), and its diameter size is 50–100 nm.

### 2.2. Synthesis of CB/WPUs

First, CB was added to 10 mL of MEK and dispersed using an ultrasonicator. Subsequently, CO and IPDI were added to the 500-mL three-necked flask. Nitrogen was passed through the flask to heat it to 75 °C. A mechanical mixer was employed to mix the content at 300 rpm. After 2 h of reaction had passed, DMPA was added for prepolymerization. When the stickiness of the mixture had increased, butanone containing CB was added to moderate the stickiness of the mixture. The WPU prepolymer was then cooled to 50 °C and neutralized using TEA for 20 min. Deionized water and ethylenediamine (EDA) were added to the mixture for 1 h to disperse the CB/WPUs (Scheme 1). The obtained CB/WPU solution was poured into a serum bottle and stored under a vacuum to defoam for 1 day. Finally, the CB/WPU solution was poured onto a Teflon plate and placed in an oven to dry. The recipe, symbols, and theoretical contents of the hard and soft segments for the CB/WPUs films are shown in Table 1.

### 2.3. Fourier Transform Infrared Spectroscopy (FT-IR)

Fourier transform infrared spectroscopy measurements were performed on a Digilab (Hopkinton, MA, USA) (model (FTS-1000)). The spectra of the samples was obtained by averaging 16 scans in a range of 4000 to 650 cm^−1^ with a resolution of 2 cm^−1^.

### 2.4. Surface Roughness Analysis

Scanning was performed using a Bruker dimension icon atomic force microscope (Billerica, MA, USA), which is generally operated in two imaging modes: tapping and contact. The tapping mode was used in this study, and the tip of the oscillation probe cantilever made only intermittent contact with the sample. Regarding the phase of the sine wave that drives the cantilever, the phase of the tip oscillation is extremely sensitive to various sample surface characteristics; therefore, the topography and phase images of a sample’s surface can be detected.

### 2.5. Thermogravimetric Analysis (TGA)

Thermogravimetric analysis was performed on a TA instrument Q-500 (New Castle, DE, USA). The samples (5–8 mg) were heated from room temperature to 700 °C under nitrogen at a rate of 10 °C/min.

### 2.6. Conductivities

The sheet resistances of the CB/WPUs were measured with a Keithley 2400 digital source meter equipped with a four-point probe. Every resistance was tested 3 times and the average value was obtained. Generally speaking, the conductive materials used in the antistatic materials can be divided into three grades. The first one is basic antistatic grade with a 10^9^~10^12^ of surface impedance. The second one is antistatic grade (static dissipation) with 10^6^~10^9^ of surface impedance. The third one is conductive grade with 10^4^~10^6^ of surface impedance.

### 2.7. Dynamic Mechanical Analysis (DMA)

Dynamic mechanical analysis was performed on DMA Q800 machine (TA Instruments, New Castle, DE, USA) at 1 Hz with a 5 μm amplitude over a temperature range of −50 to 50 °C at a heating rate of 3 °C/min. Specimens with dimensions of 35.6 × 12.7 × 2 mm^3^ were used in these tests. The Tg was taken as the peak temperature of the glass transition region in the tan δ curve.

### 2.8. Stress–Strain Testing

Tensile strength and elongation at break were measured using a universal testing machine (model CY-6040A8, Chun Yen Testing Machine Co., Ltd., Taichung, Taiwan). Testing was conducted with ASTM D638. The dimension of the film specimen was 45 mm × 8 mm × 0.2 mm.

### 2.9. Hydrolytic Degradation Tests

Hydrolytic degradation evaluation of the specimens was conducted in a 1% aqueous NaOH solution [30,31]. In order to accelerate the tests, all samples were tested at 45 °C. The specimens with dimensions of 2 × 2 cm^2^ were tested for various days, washed with distilled water, and dried completely in a vacuum oven at 70 °C for 3 h. The degree of degradation was determined from the weight loss in Equation (1):(1)$$\mathrm{Weight}\;\mathrm{loss}=\frac{W_{0}-W_{t}}{W_{0}}$$
where *W*_0_ is the dry weight before degradation, and *W_t_* is the dry weight at time *t*.

### 2.10. Morphology Analysis

Morphology of the specimens after hydrolytic degradation was observed by using a high resolution field-emission scanning electron microscope (FESEM), model JSM-6500F(JEOL, Tokyo, Japan). Specimens of 2 × 2 cm^2^ were fixed on a sample holder using conductive adhesive tape and were then coated with a thin layer of platinum to improve image resolution. The samples were photographed with 1.00 K magnification

## 3. Results and Discussion

### 3.1. Fourier Transform Infrared Spectroscopy (FT-IR)

Figure 2a displays the FTIR analysis results and reveals that the curve of the CB/WPUs is within 4000–650 cm^−1^. The FTIR curve contains the characteristic peak of PU (–COONH–). The stretching vibration peaks of the –NH and –OH groups are located at 3100–3550 cm^−1^, whereas the stretching vibration peak of CH_2_ is located at 2800–3000 cm^−1^ (the asymmetric and symmetric stretching vibration peaks of CH_2_ are located at 2923 and 2856 cm^−1^, respectively). The stretching vibration peak of C=O is located at 1600–1800 cm^−1^ and represents the ordered H-bonded carbonyl groups (C=O_order_), disordered H-bonded carbonyl groups (C=O_disorder_), and free carbonyl groups (C=O_free_). The stretching vibration peaks for amide II (δN–H + νC–N + νC–C) and III (νC–N, N–H bending, and C–Cα) are located at 1529 and 1234 cm^−1^, respectively. These results are consistent with those in other reports [6,32]. This study conducted curve fitting of the C=O curve section (1620–1740 cm^−1^) to determine changes in the number of H-bonds and reveal the influence of adding CB to the WPU on H-bond interactions (Figure 2b). The results revealed that the stretching peaks of C=O_order_, C=O_disorder_, and C=O_free_ are located at 1644, 1703, and 1738 cm^−1^, respectively. After calculating the curve area of C=O_order_, C=O_disorder_, and C=O_free_, the percentage of total H-bond content within the C=O functional group of each CB/WPU was determined (Table 2). The calculations indicated that the C=O functional group comprised 79.96%, 88.27%, 92.28%, and 84.90% of the total H-bond content in WPU, CB/WPU-01, CB/WPU-02, and CB/WPU-03, respectively. The original C=O comprised 79.96% of the total H-bond content of the WPU. These H-bonds were located in the hard section of the C=O⋯H–N. When the amount of CB added was 0–2 wt.%, higher amounts of CB resulted in more total H-bonds in the C=O functional group, with the maximum difference being 12.32%. These H-bonds were located in the C=O⋯H–O section between WPU and CB. When 3 wt.% CB was added, the total number of H-bonds was lower. This was possibly due to severe clustering of CB that prevented the formation of H-bonds.

### 3.2. Surface Roughness and Electrical Resistance Analysis

The hydrogen bond is one of the key factors promoting the microphase separation of PU. So on the whole, it can be said that the hydrogen bond influences the surface roughness and also influences the distribution of microphase separation at the same time. The left and right parts of Figure 3 display the three-dimensional morphology and phases of the CB/WPUs. The average surface roughness of WPU, CB/WPU-01, CB/WPU-02, and CB/WPU-03 is 1.58, 2.07, 4.32, and 2.84 nm, respectively. Up to CB addition of 2 wt.%, an increase in the CB added caused an increase in the surface roughness of the WPU. However, adding an excessive amount of CB resulted in lower average surface roughness due to the H-bond interaction produced by the –OH group of CB and –C=O group of PU. The FTIR results revealed that CB/WPU-02 had the most H-bonds, causing an unstable film-forming process and increasing the average surface roughness. The phase figure indicates that WPU had similar microphase separation conditions as PU. When 1 wt.% of CB was added, the number of surface hard segments was increased. The hard segments (i.e., the white dots in Figure 3) showed the presence of CB. Because in the phase diagram of AFM, it is able to identify the soft or hard degree of the film, and when the values are higher (white dots), it represents the film is harder. The microphase separation in PU can promote the separation of soft segment and hard segment. The hard segment belongs to the harder area in the film. So the white dots represent the hard segment (IPDI), but they can also represent CB (because CB is relatively harder compared to the film). This indicates that CB promotes microphase separation. When 2 wt.% of CB was added, the hard segments began to cluster and the soft segments (i.e., darker area) were evenly distributed. This result indicated that the degree of microphase separation was greater in CB/WPU-02 than in CB/WPU-01. The phase figure of CB/WPU-03 indicates that when the amount of CB added was 3 wt.%, the white dots were clustered. Microphase separation in PU is caused by the different polarity of hard and soft segments. Additionally, hard segment clustering is caused by H-bond interactions between hard segments. Because the CB/WPU-02 film exhibited the most H-bond interactions, it had the greatest degree of microphase separation.

The amount of CB substantially influences the conductivity of CB/WPUs. The resistive properties of the CB/WPUs are displayed in Table 3. The standard WPU did not exhibit conductivity. When 1 wt.% CB was added, the resistive value was 2.01 × 10^6^, indicating that the addition of 1 wt.% CB gave the WPU static conductivity. CB/WPU-02 and CB/WPU-03 had resistive values of 8.41 × 10^5^ and 3.99 × 10^4^, respectively. This was because more CB was present in the WPU and indicates that all samples had static conductivity.

### 3.3. Thermal Properties

Figure 4 displays the TGA curves of WPUs containing different amounts of CB. Table 2 records the following TGA values of the samples: *T*_onset_, *T*_50_, and 700 °C residue. The results indicated that the *T*_onset_ of WPU, CB/WPU-01, CB/WPU-02, and CB/WPU-03 were 290.49 °C, 294.89 °C, 297.13 °C, and 299.58 °C, respectively. Additionally, the T_50_ of WPU, CB/WPU-01, CB/WPU-02, and CB/WPU-03 were 337.05 °C, 344.55 °C, 347.21 °C, and 348.43 °C. Thus, when 3 wt.% CB was added to the WPU, the temperature required to reach *T*_onset_ or *T*_50_ was increased by 10 °C. This was because CB has greater thermostability than WPU. Therefore, the thermostability of WPU was increased by adding more CB. Furthermore, the 700 °C residue masses of WPU, CB/WPU-01, CB/WPU-02, and CB/WPU-03 were 1.81%, 2.24%, 2.30%, and 2.45%, respectively. The increase of residue mass in CB/WPUs with CB content again indicates that more CB has remained in WPU.

### 3.4. Dynamic Mechanical Analysis (DMA)

Figure 5 displays the tanδ of the WPUs obtained when different amounts of CB were added. The dynamic glass transition temperature is denoted *T*_gd_. Table 3 lists the maximum *T*_gd_ and tanδ for the different CB/WPUs. The *T*_gd_ of the standard WPU was 52.66 °C. Adding 2 wt.% CB to WPU resulted in a higher *T*_gd_ of 59.41 °C. This was possibly due to H-bond interactions between the –OH groups in CB and C=O groups in WPU. This H-bond interaction prevents WPU molecular chains from moving, causing a higher *T*_gd_. However, when 3 wt.% CB was added, the *T*_gd_ was lower. This may have been caused by the extensive clustering of CB, which restrained the –OH groups of CB and prevented them from forming H-bonds with the C=O groups, as was demonstrated in the FTIR analysis. Additionally, the maximum tanδ of WPU, CB/WPU-01, CB/WPU-02, and CB/WPU-03 was 0.403, 0.400, 0.398, and 0.384, respectively. The tanδ is obtained from the loss modulus divided by the storage module. So, lower tanδ value means harder membrane. Thus, the maximum tanδ decreased with an increase of CB. This was because the addition of CB increased the hardness of the CB/WPUs, reducing the flexibility and increasing the stickiness of the WPUs.

### 3.5. Tensile Properties

Figure 6 displays the stress–strain curve of the various CB/WPUs. Table 4 records the maximum tensile strength, breaking strain, and Young’s modulus of each CB/WPU. The maximum tensile strength of WPU, CB/WPU-01, CB/WPU-02, and CB/WPU-03 was 9.7, 12.9, 13.8, and 10.4 MPa, respectively, whereas their breaking strains were 123%, 108%, 69%, and 60%. CB/WPU-02 thus had the greatest tensile strength. Possible reasons for this include the dispersibility of CB when 2 wt.% CB was added to WPU and the H-bond interactions between CB and WPU. In these two situations, the degree of microphase separation was increased. This microphase separation result was also shown in the AFM results. When an excessive amount of CB (i.e., 3 wt.%) was added, the CB in the WPU was severely clustered. This clustering increased the stress concentration point of the WPU and caused more defects in the material, reducing the maximum tensile strength and breaking strain. The Young’s modulus of WPU, CB/WPU-01, CB/WPU-02, and CB/WPU-03 was 0.85, 1.39, 3.64, and 1.77 MPa, respectively. CB/WPU-02 thus exhibited the greatest Young’s modulus. This result is consistent with the aforementioned description.

### 3.6. Hydrolytic Degradation

This study sought to develop an environmentally friendly and degradable material. Figure 7 displays the mass loss of the CB/WPUs after degradation in 1 wt.% NaOH. The mass loss of all samples exceeded 15% after 12 h of degradation. After 16 h of degradation, most of the polymer structure was fractured, causing difficulties in sampling. CB/WPU-02 exhibited the lowest degradation speed. This was because the CB/WPU-02 had the most H-bond interactions. When the amount of CB added was 3 wt.%, the CB was severely clustered within the WPU, increasing the number of defects in the material. This enabled 1 wt.% NaOH to enter the CB/WPU-03 film more easily and caused the CB/WPU-03 to lose more mass.

### 3.7. SEM Morphology Analysis

Figure 8 displays the SEM surface topology of CB/WPUs after degradation in 1 wt.% NaOH for different amounts of time. After 12 h, numerous holes had developed on the WPU sample containing no CB, whereas CB/WPU-01 had considerably fewer holes. This was because the H-bond interactions between CB and WPU prevented the 1 wt.% NaOH solution from infiltrating the WPU film. The surface topography of CB/WPU-02 is rougher than the other samples. This was possibly because more H-bond interactions were causing the greatest degree of microphase separation within the CB/WPU-02 (such as AFM). The high degree of microphase separation promoted the clustering of degradable casotor oil segments, so after CB/WPU-02 was immersed in 1 wt.% NaOH solution, the clustered soft segments on CB/WPU-02 surface were degraded and the shape was changed. After being degraded for 12 h, large cavities formed on CB/WPU-03. This was because of severe CB clustering in the WPU, which caused numerous defects in the CB/WPU-03 film. In summary, from the degradation loss rate and SEM, it was known that CB/WPU-02 had less degradation loss rate and no formation of cavities, and this was possibly because the H-bond interactions inhibited 1 wt.% NaOH, permeating into CB/WPUs film.

## 4. Conclusions

This study successfully synthesized an environmentally friendly vegetable-oil-based PU. FTIR analyses revealed that CB/WPU-02 had the most H-bond interactions. The AFM and conductivity test results revealed that 2 wt.% CB was the optimal amount of CB to add to promote microphase separation. This was due to CB microclustering when 2 wt.% CB was employed. The CB microclustering and H-bond interactions caused CB/WPU-02 to exhibit the highest degree of microphase separation. Additionally, TGA revealed that greater amounts of CB in the WPU resulted in higher initial degradation temperatures. The DMA and tensile test revealed that the CB/WPU-02 film had the highest *T*_gd_ and most favorable mechanical properties. When the amount of CB added was 3 wt.%, the CB clustered severely and numerous defects were created. The degradation experiment and SEM results revealed that CB/WPU-02 exhibited the greatest degradation stability of all samples. NaOH was found to not easily infiltrate the CB/WPU-02 film, causing the degradation process to commence from the film surface. In summary, when 1 wt.% of CB is added (CB/WPU-01), it has already become the static conductive material (such as Table 3). So the cost has already been reduced, it is much more suitable to be used in the application of operation room and weapon storehouse. As for the raise of benefit, 2 wt.% of CB addition (CB/WPU-02) will be the most appropriate addition level.

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
