# Peer review of "Synthetic Environmentally Friendly Castor Oil Based-Polyurethane with Carbon Black as a Microphase Separation Promoter"

_polymers, 2019, doi:10.3390/polym11081333_

Round 1

Reviewer 1 Report

The authors presented an interesting topic on the synthetic and fabrication of environmentally friendly polyurethane composites. The effects of the carbon black content on the microphase separation, conductivity, thermal and mechanical properties, and degradation of the composites were also investigated. However, there are questions or concerns about the manuscript and the English languages needs to be proof-read.  

The following are the detailed comments:

1.    The authors claimed that CB is a microphase separation promoter, is this a target of adding CB or the by-product of the adding CB? What is the reason to achieve microphase separation? Is it due to the mechanical property improvement? Is the mechanical properties improvement due to microphase separation or the reinforcing effect of the filler? Or both?

2.    Line 19, is it true that CB is not necessarily used to create the PU?

3.    Line 71, what does “conductive and static conductive materials” mean? What are the differences between “conductive” and “static conductive”?

4.    Line 75. It seems that Figure 1 does not represent the detailed process described here, but Scheme 1.

5.    Line 88, could author give the details of the producer of these products?

6.    Line 95, what is “a suitable amount”?

7.    Line 158, Format of the reference.

8.    Figure 2, which line the arrow are pointing to?

9.    Table 2, what does “HB” stand for?

10.  Section 3.2., how was the surface roughness related to the microphase separation and distribution of the microphase separation?

11.  Line 188, “Figure 4” should be replace with “Figure 3”.

12.  Line 188, Is the white dots caused by clustering of CB or CB-promoted IPDI hard segment?

13.  Line 190, where are the black dots in the pictures? Could some be articled to show?

14.  Figure 3, why is the picture for CB/WPU-02 so different from the others?

15.  Line 217, how does the residue at 700 °C align with the 1, 2, 3 wt% CB? maybe it is just because of the additional CB leading to high residue?

16.  Line 229, is the number statistically different?

17.  Line 231, How does added CB increase the flexibility of the WPU?

18.  Section 3.6, is there any test standard?

19.  Line 253, the concentration of sodium hydroxide used was 3 wt % in the experimental section.

20.  Line 267, Should author use more accurate word instead of "distinctive"? it could mean both ways, better or worse.

21.  Line 270, should all the attack start from the surface instead of from the inside?

22.  Line 270-271, Where was the conclusion drawn from? From the pics, it seems the CB/WPU-03 did not show degradation and even less severe than CB/WPU-02. It looks much better than CB/WPU-01 for my viewpoint.

23.  Line 290-292, how did the authors come up with this conclusions? It is misleading to reach the summary without any comparison or systematic study.

Author Response

Manuscript ID:polymers-557712

Title: Synthetic Environmentally Friendly Vegetable Oil Based-Polyurethane with Carbon Black as a Microphase Separation Promoter

Comments 1.

The authors presented an interesting topic on the synthetic and fabrication of environmentally friendly polyurethane composites. The effects of the carbon black content on the microphase separation, conductivity, thermal and mechanical properties, and degradation of the composites were also investigated. However, there are questions or concerns about the manuscript and the English languages needs to be proof-read.  

1.The authors claimed that CB is a microphase separation promoter, is this a target of adding CB or the by-product of the adding CB? What is the reason to achieve microphase separation? Is it due to the mechanical property improvement? Is the mechanical properties improvement due to microphase separation or the reinforcing effect of the filler? Or both?

Ans: As for polyurethane (PU), the more microphase separation can increase the mechanical properties of PU (Line 44-46 in the text). “The formation of microphase separation is mainly arisen from the generation of hydrogen bond between the -NH group on urethane group (-COONH-) hard segment and C=O group, which promotes the aggregation of hard segment”. It has already been added to Line 46-48. The -OH on the surface of every CB can form hydrogen bond with C=O on urethane group (COONH). These radical groups can attract the hard segments to around CB to increase the microphase separation.” (It has already been added to Line 72-73).

Line 19, is it true that CB is not necessarily used to create the PU?

Ans: CB only represented the existence of additive in this text. So there is WPU without CB in the study samples.

Line 71, what does “conductive and static conductive materials” mean? What are the differences between “conductive” and “static conductive”?

Ans: The word of conductive in Line 75 has already been deleted. “Generally speaking, the conductive materials used in the antistatic materials can be divided into three grades. The first one is basic antistatic grade with 109~1012 of surface impedance. The second one is antistatic grade (static dissipation) with 106~109 of surface impedance. The third one is conductive grade with 104~106 of surface impedance.” The statement has been added in line 131-134 of the revised manuscript.

Line 75. It seems that Figure 1 does not represent the detailed process described here, but Scheme 1.

Ans: In Line 79, Figure 1 has already been corrected as Scheme 1, and Figure 1 shows cycle of environmental of CB/WPUs has been added to Line 80.

Line 88, could author give the details of the producer of these products?

Ans: The detailed information of manufacturer has already been supplemented to Section 2.1.

6.Line 95, what is “a suitable amount”?

Ans: Line 101 has already been revised to “CB was added to 10 mL of MEK…”.

Line 158, Format of the reference.

Ans: In Line 168, [6,[30] has already been corrected to [6,32].

Figure 2, which line the arrow are pointing to?

Ans: In order to prevent the figure from being too disorder, so we choose to disclose 3 peak values separately, and we supplemented (C =O order as blue line, C =O disorder as green line, C =O free as pink line) to Figure 2 title in Line186.

Table 2, what does “HB” stand for?

Ans: In Table 2, “HB content” has already been corrected to “H-bond content”.

Section 3.2., how was the surface roughness related to the microphase separation and distribution of the microphase separation?

Ans: In Line195-196, the change of the surface roughness is mainly influenced by the hydrogen bond (can be verified by FTIR). “The hydrogen bond is one of the key factors promoting the microphase separation of PU. So on the whole it can be say that the hydrogen bond influences the surface roughness and also influences the distribution of microphase separation at the same time” as explained in Line 189-191.

Line 188, “Figure 4” should be replace with “Figure 3”.

Ans: In Line 201, “Figure4” has already been corrected to “Figure 3”.

Line 188, Is the white dots caused by clustering of CB or CB-promoted IPDI hard segment?

Ans: Because in the phase diagram of AFM, it is able to identify the soft or hard degree of the film, and when the values are higher (white dots), it represents the film is harder. The microphase separation in PU can promote the separation of soft segment and hard segment. The hard segment belongs to the harder area in the film. So the white dots mean the hard segment (IPDI), but they can also represent CB (because CB is relatively harder compared to the film). This has already been supplemented to Line 201-206.

Line 190, where are the black dots in the pictures? Could some be articled to show?

Ans: “The black dots” originally just want to correspond to the white dots, correctly speaking, they mean the darker area. It has already been corrected in Line 207.

Figure 3, why is the picture for CB/WPU-02 so different from the others?

Ans:At present, from FTIR it is known that CB/WPU-02 has the strongest hydrogen bond. From the mechanical property, it is known that the mechanical property of CB/WPU-03 is reduced. So we infer in CB/WPU-02, CB produces the best microphase separation in WPU and causes the change of Phase diagram in CB/WPU-02, as explained in Line 206-208.

Line 217, how does the residue at 700 °C align with the 1, 2, 3 wt% CB? maybe it is just because of the additional CB leading to high residue?

Ans:In Line 236, “CB increases the thermostability of WPU.” has already been corrected to “more CB is remained in WPU. "

Line 229, is the number statistically different?

Ans:It is tanδ value in Line 247. The tanδ is obtained from the loss modulus divided by the storage module. So lower tanδ value means harder membrane. The statement has been added in line 247-248 of the revised manuscript.

Line 231, How does added CB increase the flexibility of the WPU?

Ans: In Line 250, “reducing the stickiness and increasing the flexibility of the WPUs.” has already been corrected to “reducing the flexibility and increasing the stickiness of the WPUs.”

Section 3.6, is there any test standard?

Ans: In the literatures, there are many different hydrolysis conditions. We “conducted the tests in accordance with references [30, 31], and in order to accelerate the tests, all samples were tested at 45 °C”. This has already been supplemented to Section 2.9.

Line 253, the concentration of sodium hydroxide used was 3 wt % in the experimental section.

Ans: The 3 wt % has already been corrected to 1 wt % in experimental section 2.9.

Line 267, Should author use more accurate word instead of "distinctive"? it could mean both ways, better or worse.

Ans: In Line 286, “distinctive” has already been corrected to “rougher”.

Line 270, should all the attack start from the surface instead of from the inside?

Ans: The prepared CB/WPU solution was CB/WPU resin dispersed in water, not soluble in water. The prepared CB/WPU film still had some water-resistance. During the hydrolytic degradation test, entire surface area of the CB/WPU film was exposed to 1wt% aqueous NaOH solution, whereas most inside area of the film was protected by the surface area. Thus the corrosion attack started from the surface area and we use SEM morphology analysis (Section 3.7) to evaluate the effect of hydrolytic degradation (Section 3.6)

Line 270-271, Where was the conclusion drawn from? From the pics, it seems the CB/WPU-03 did not show degradation and even less severe than CB/WPU-02. It looks much better than CB/WPU-01 for my viewpoint.

Ans: We tried to take the photographs for the degraded CB/WPU-03 and CB/WPU-02 samples again, and corrected them in Figure 8. “This was possibly because more H-bond interactions causing the greatest degree of microphase separation within the CB/WPU-02 (such as AFM). The high degree of microphase separation promote the clustering of degradable casotor oil segments, so after CB/WPU-02 was immersed in 1 wt. % NaOH solution, the clustered soft segments on CB/WPU-02 surface were degraded and the shape was changed. After degraded for 12hr, large cavities were formed on CB/WPU-03”. This has already been corrected in Line 286-290. “In summary, from the degradation loss rate and SEM, it was known that CB/WPU-02 had less degradation loss rate and no formation of cavities, this was possibly because the H-bond interactions inhibited 1 wt. % NaOH permeating into CB/WPUs film.” This has already been corrected in Line 292-295.

Line 290-292, how did the authors come up with this conclusions? It is misleading to reach the summary without any comparison or systematic study.

Ans: “When 1wt% of CB is added (CB/WPU-01), it has already become the static conductive material (such as Table 3). So the cost has already been reduced and much suitable to be used in the application of operation room and weapon storehouse. As for the raise of benefit, 2wt% of CB addition (CB/WPU-02) will be the most appropriate addition level.” It has already been corrected in Line 311-315.

Reviewer 2 Report

This manuscript deals with the investigation of Carbon Black as microphase separation promoter in environmentally friendly water-polyurethane. Structural, mechanical and morphological characterizations are reported together with preliminary test on the chemical stability under hydrolytic stress. The results are very interesting and clearly exposed. However, there is a critical point that deserve to be clarified. From the discussion clearly emerges that the nature and the number of H-bond interactions between Carbon Black and Water polyurethane strongly affect the overall properties of the resulting polymer. In this regard, the increased number of HB has been demonstrated by FTIR investigation. Unfortunately, I believe that FTIR analysis in transmittance mode has low suitability for quantitative estimation of hydrogen bond content. Furthermore, also in adsorbance mode, variations in sample preparation could also affect the results.

Accordingly I’d suggest to address and clarify this point and other few issues (listed below) prior to accept the manuscript.

Comments:

1.      Page 4, row 105: the scheme reports the synthesis of Castor oil based-WPU, please check the caption.

2.      Page 9, rows 229-231: the sentences: “Thus, the maximum tan δ decreased with an increase of CB. This was because addition of CB increased the hardness of the CB/WPUs, reducing the stickiness and increasing the flexibility of the WPUs”. Lower tan δ values indicates reduced damping capacity rather than higher flexibility. Please correct the phrase.

3.      Page 12, rows 290-292:  the sentences: “In summary, the environmentally friendly CB/WPU-01 is suitable for use in surgery rooms and arsenals, whereas the environmentally friendly CB/WPU-02 is applicable as an industrial coating”. From the manuscripts, are not clear the reasons why CB/WPU-01 betters fit the needs of surgery rooms and arsenals applications while CB/WPU-02 should be applicable as industrial coating. Please explain it.

Author Response

Comments 2.

This manuscript deals with the investigation of Carbon Black as microphase separation promoter in environmentally friendly water-polyurethane. Structural, mechanical and morphological characterizations are reported together with preliminary test on the chemical stability under hydrolytic stress. The results are very interesting and clearly exposed. However, there is a critical point that deserve to be clarified. From the discussion clearly emerges that the nature and the number of H-bond interactions between Carbon Black and Water polyurethane strongly affect the overall properties of the resulting polymer. In this regard, the increased number of HB has been demonstrated by FTIR investigation. Unfortunately, I believe that FTIR analysis in transmittance mode has low suitability for quantitative estimation of hydrogen bond content. Furthermore, also in adsorbance mode, variations in sample preparation could also affect the results. Accordingly I’d suggest to address and clarify this point and other few issues (listed below) prior to accept the manuscript.

Ans: We try to revise the DATA in the same group to the absorption mode in the software (the machine is Digilab (Hopkinton, MA, USA) (model (FTS-1000) ). The relation of absorbance and transmittance can be shown in the following equations:

T%=Sample/Background×100

A=log10(1/T)

Where A is the absorbance and T is the transmittance. After calculating, obtain similarity of diagram and transmitting (such as Line 183), and carry on the curve fitting by the same step to get similar result and revise the text (such as Line 174, 175, 178, 184 and Table 2).

Comments:

Page 4, row 105: the scheme reports the synthesis of Castor oil based-WPU, please check the caption.

Ans: In Line 112, the title of the Scheme 1 have been revised of the revised manuscript. (Formula for the castor oil based-WPU)

Page 9, rows 229-231: the sentences: “Thus, the maximum tan δ decreased with an increase of CB. This was because addition of CB increased the hardness of the CB/WPUs, reducing the stickiness and increasing the flexibility of the WPUs”. Lower tan δ values indicates reduced damping capacity rather than higher flexibility. Please correct the phrase.

Ans: In Line 250, “Reducing the stickiness and increasing the flexibility of the WPUs.” has already been corrected to “reducing the flexibility and increasing the stickiness of the WPUs.”

Page 12, rows 290-292:  the sentences: “In summary, the environmentally friendly CB/WPU-01 is suitable for use in surgery rooms and arsenals, whereas the environmentally friendly CB/WPU-02 is applicable as an industrial coating”. From the manuscripts, are not clear the reasons why CB/WPU-01 betters fit the needs of surgery rooms and arsenals applications while CB/WPU-02 should be applicable as industrial coating. Please explain it.

Ans: “When 1wt% of CB is added (CB/WPU-01), it has already become the static conductive material (such as Table 3). So the cost has already been reduced and much suitable to be used in the application of operation room and weapon storehouse. As for the raise of benefit, 2wt% of CB addition (CB/WPU-02) will be the most appropriate addition level.” It has already been corrected in Line 311-315.

Round 2

Reviewer 2 Report

The revised paper is well-done and the results are now clearly exposed.
The manuscript is worthily to be published in the present form.